# Nanometric flow and earthquake instability

Hongyu Sun [1 ✉] & Matej Pec [1 ✉]

Fault zones accommodate relative motion between tectonic blocks and control earthquake nucleation. Nanocrystalline fault rocks are ubiquitous in "principal slip zones" indicating that these materials are determining fault stability. However, the rheology of nanocrystalline fault rocks remains poorly constrained. Here, we show that such fault rocks are an order of magnitude weaker than their microcrystalline counterparts when deformed at identical experimental conditions. Weakening of the fault rocks is hence intrinsic, it occurs once nanocrystalline layers form. However, it is difficult to produce "rate weakening" behavior due to the low measured stress exponent, $n$, of $1.3 \pm 0.4$ and the low activation energy, $Q$, of $16,000 \pm 14,000$ J/mol implying that the material will be strongly "rate strengthening" with a weak temperature sensitivity. Failure of the fault zone nevertheless occurs once these weak layers coalesce in a kinematically favored network. This type of instability is distinct from the frictional instability used to describe crustal earthquakes.

[1] Department of Earth, Atmospheric and Planetary Sciences, Massachusetts Institute of Technology, 77 Massachusetts Ave, Cambridge, MA 02139, USA.
✉email: hongyus@mit.edu; mpec@mit.edu

Fault zones are complex geological structures where deformation partitions and localizes into high-strain and low-strain domains over a wide range of length scales[1,2]. The interactions and feedbacks between the domains ultimately determine the mode of fault slip, from earthquakes, slow slip to viscous creep[2]. Localization in the high-strain domains is a prerequisite for numerous weakening mechanisms such as thermal pressurization[3,4] and shear heating[5,6] that may induce weakening and may lead to an earthquake instability[7]. Furthermore, the material where strain localizes becomes profoundly transformed; high local work input induced by localization promotes comminution[8], metamorphic reactions[9–11], microstructural transformations[12,13], phase transitions[14,15] and melting[16–18] that can locally weaken the fault rocks over a broad range of pressure, temperature (P-T) and strain rate conditions and produce nanocrystalline to amorphous materials in nature[19–21] as well as in experiments[16,22–25].

The rheology of these fine-grained fault rocks in zones of extreme localization is therefore critical for understanding fault slip[14,26,27], yet it remains poorly constrained. Because of a lack of appropriate experimental studies, current models of nanocrystalline fault rock rheology rely on microstructural inferences and extrapolations of flow laws derived for microcrystalline materials[27]. These extrapolations, however, result in large uncertainties since nanocrystalline, surface dominated, materials typically have physical and mechanical properties that are distinct from microcrystalline, volume dominated, materials[28].

Nanocrystalline fault rocks typically form compact zones with little to no porosity, composed of crystals (1's–10's of nm) to crystal aggregates (100's of nm) that are sometimes embedded in an amorphous matrix[12,14,16,22,23,26,29,30]. The small grain size and associated short diffusional distances in combination with highly disordered lattices make diffusion competitive at conditions where cataclastic flow typically dominates[26,31]. Numerous lines of evidence suggest that the nanocrystalline fault rocks are weaker than the surrounding, coarser-grained, material and flow even at low ambient temperatures[12,26,32]. The exact cause of this weakening is, however, difficult to pin-point as several processes operate in parallel and could be responsible.

The principal problem in determining the rheology of nanocrystalline fault rocks is in isolating their intrinsic properties from the mechanical signal which is dominated by coarser-grained material in low velocity, small-displacement experiments[22,24,26]. In high velocity, large-displacement experiments, temperature transients make it difficult to untangle the effect of heating from the effects of nanomaterial formation[23,32–35]. At what stage of fault slip did the material form? Did the material reach high temperatures during slip? Is the nanocrystalline material the cause or the consequence of failure?

To circumvent these problems, we have produced granitoid nanomaterials in bulk by high-energy ball milling (details in "Methods") and tested their rheological properties using a solid medium deformation apparatus under controlled P-T conditions corresponding to the base of the seismogenic layer where many large earthquakes nucleate.

## Results

### Mechanical properties of granitoid nanometric fault rocks.
The nanomaterials are prepared from crushed granitoid gouge of the same origin as used in the previous studies[12,30,31]. Examination under scanning electron microscope using back-scattered electrons (SEM-BSE; Supplementary Fig. 1) shows that all minerals are thoroughly mixed due to ball milling, so it is impossible to distinguish individual grains. The material is cohesive and of uniform gray in BSE images suggesting a uniform chemical composition. Initial grain size determined by laser particle sizer is $0.01 \leq d \leq 1\,\mu m$, with a median of ~100 nm. More properties of the starting materials and a detailed explanation of the experimental procedures are described in "Methods".

We compare the grain-size dependent strength of the fault rocks using constant-displacement-rate experiments (Methods, Fig. 1a). Note that microcrystalline experiments develop patches of nanocrystalline material with increasing strain that interconnect at peak stress and allow for failure[24,30]. All experiments were conducted in a general shear geometry at temperatures, $T = 200$, 300 and 500 °C and confining pressures, $P_c = 500$ MPa at a constant displacement rate of ~$10^{-3}$ mm s$^{-1}$, corresponding to a shear strain rate, $\dot{\gamma}$, of ~$10^{-3}$ s$^{-1}$. As shown in Fig. 1, the strength of the granitoid fault rocks is clearly decreased due to the reduced grain size. Furthermore, the material with starting grain size of $\leq 200\,\mu m$ showed an abrupt stress-drop at 300 °C, a laboratory equivalent to an earthquake[36], but the nanomaterial creeps with no loss of strength at the same temperature. With the increase of temperature, the maximum shear stress ($\tau$), apparent viscosity ($\eta = \tau/\dot{\gamma}$), and friction coefficient ($\mu = \tau/\sigma_n$, where $\sigma_n$ is the normal stress) are all decreased for both grain sizes, indicating that the materials become weaker with increasing temperature (Fig. 1b; Table 1). The nanocrystalline fault rocks at 200 °C and 300 °C both show strain hardening after the yield point. The hardening rate decreases with increasing temperature. At 500 °C, both the microcrystalline as well as the nanocrystalline materials continue to deform at approximately constant stress implying that the rocks have reached steady-state flow. Flow in the microcrystalline experiment is accommodated by ≈20 vol% of nanocrystalline material that is produced during shearing[30]. Only for the nanocrystalline fault rocks, the differential stress, $\Delta\bar{\sigma}$, is just below the Goetze criterion ($\Delta\bar{\sigma} \leq P_c$).

To further constrain the rheology of the nanocrystalline fault rocks, we quantitatively investigate the stress exponent, $n$ (Fig. 1c) using stress-stepping experiments and estimate the activation energy, Q (Fig. 1d) using the constant-displacement-rate experiments (Methods, Supplementary Fig. 3, Table 1). The stress-stepping experiments indicate that the nanocrystalline fault rocks exhibit a near-linear stress–strain rate relationship with a stress exponent, $n = 1.3 \pm 0.4$, for temperatures ranging from 300 to 500 °C. Constant-displacement-rate experiments provide an estimation of the activation energy, $Q = 16,000 \pm 14,000$ J/mol, for stress exponent of 1.3. Hence, the flow law for granitoid nanometric fault rocks is determined as

$$\dot{\varepsilon} = 3.69 \cdot 10^{-6} \bar{\sigma}^{1.3} exp\left(\frac{-16000}{RT}\right) \qquad (1)$$

where $\bar{\sigma}$ is equivalent stress in MPa, $\dot{\varepsilon}$, equivalent strain rate in $s^{-1}$, $R$, gas constant, of 8.314 J/(mol·K), $T$, absolute temperature in Kelvin. The unit of the activation energy in the exponential term is in J/mol. Further verification of the flow law shows that the calculated strain rates compare well to the measured strain rates in the constant displacement rate experiments (Supplementary Fig. 4, Supplementary Table 1). Hence, it appears that nanometric fault rocks can dominantly deform by diffusion creep—as suggested by the characteristic stress exponent of ~1[37] - even at low temperatures and fast experimental strain rates. While the stress exponent is consistent with microcrystalline materials deforming dominantly by diffusion creep, the activation energy is substantially lower in the nanocrystalline fault rocks. Low activation energies are expected for nanomaterials due to the high surface-to-volume ratio; it requires less energy to free an atom from the surface since the atoms in nanomaterials have on average fewer bonds with neighboring atoms compared to coarser-grained materials[38,39].

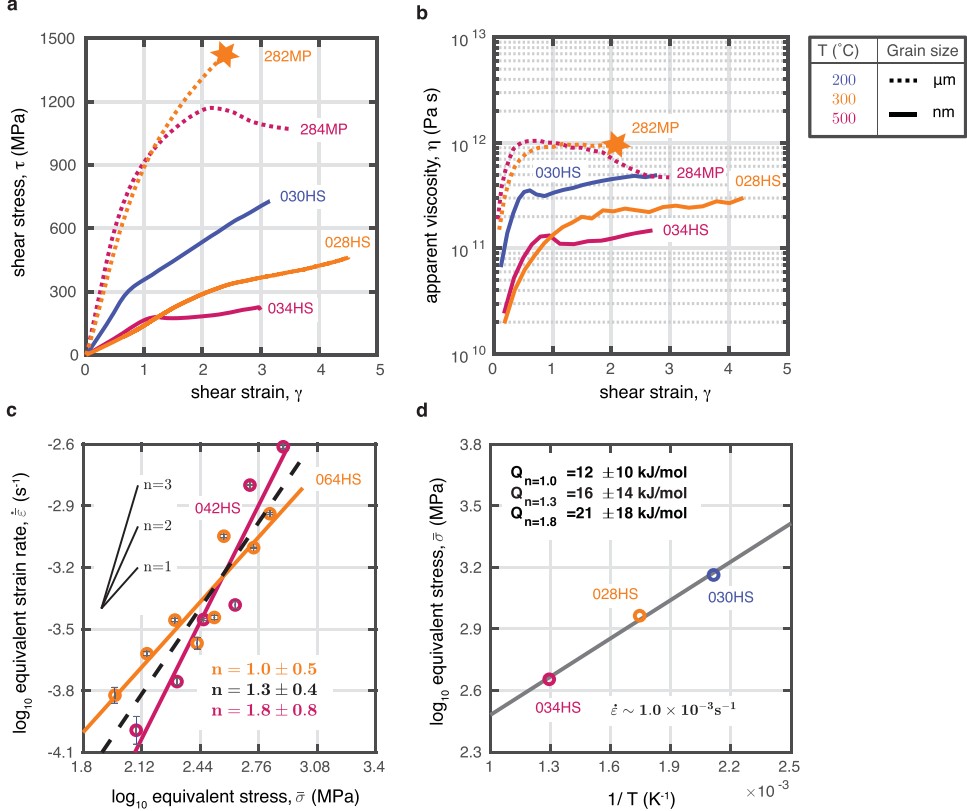

**Fig. 1 Mechanical data. a** Comparison of the shear strength between nanocrystalline fault rocks, ~0.1 μm, and microcrystalline fault rocks, ≤200 μm. The nanocrystalline fault rocks are about 1 GPa weaker than the microcrystalline ones. **b** Nanocrystalline fault rocks have about an order of magnitude lower apparent viscosity than the microcrystalline fault rocks at same T. **c** Determination of the stress exponent, $n = 1.3 \pm 0.4$. The blue error bars on the plot represent the standard deviation of measurements. The stress exponent is estimated by fitting to data from both experiments, given that the activation energy of the material is low. **d** Arrhenius plot determining the activation energy using the constant-displacement-rate experiments. For details of conversion from shear to equivalent stresses and strains see the "Methods" section.

At all studied temperatures, the nanocrystalline material deformed in a stable manner without any abrupt stress drops (Fig. 1a).

**Microstructures of nanocrystalline fault rocks**. To constrain active deformation mechanisms, we studied the resulting microstructures using high-resolution SEM and polarized light microscopy. At low temperatures of 200–300 °C, Riedel shear fractures are ubiquitous with distinctive geometric relationships typical for a shear zone (Figs. 2 and 3). The $R_1$ Riedel shear fractures make a low angle of 160° with the shear zone boundary and show the same sense of slip. $R_2$-shear fractures are found at an angle of 65–85° with respect to the shear zone boundary. The cross-cutting relationships between $R_1$-shears and $R_2$-shears indicate that the $R_2$-shears develop after the establishment of $R_1$-shear fractures. At 300 °C, optical anisotropy gets more pronounced and kink bands are visible as alteration of yellow/purple and blue along the $R_2$ Riedel shear orientation (Fig. 2c). These kink bands nucleate at the extremities of the forcing blocks, migrate inwards with increasing strain (Figs. 2c, 3c, d and Supplementary Fig. 7e, 7f), and suggest that the material is mechanically anisotropic. Porosity due to opening of $R_1$ and $R_2$ fractures indicates bulk dilatancy at low temperatures (200 and 300 °C) coinciding with strain hardening behaviors and differential stresses above the Goteze criterion (Fig. 1a).

In contrast, fractures are much less common when the material is deformed at 500 °C in agreement with the mechanical steady-state and stresses below the Goetze criterion (Fig. 1a). Almost all the visible fractures are unloading cracks with some $R_2$-oriented

fractures aligned "en-echelon" in the $R_1$ orientation (Fig. 2d). Optical anisotropy is further strengthened as evidenced by more saturated interference colors (Fig. 2d). Flow features, such as smeared out domains of slightly different z-contrast, dominate suggesting that the deformation is continuous (Figs. 2d, 3e and Supplementary Fig. 7). These observations, together with the mechanical data, suggest that the material deforms as a volume-conserving viscous fluid at 500 °C.

Transmission electron microscopy (TEM) confirms that the material at 500 °C is compact with no porosity and composed of crystals with a mean grain size of 46 nm (Fig. 4). Silicate grains are rounded (aspect ratio, $b/a \approx 0.9$) with no strong shape preferred orientation (SPO) visible in a section parallel to their flow direction, whereas micas show strong SPO ($b/a \approx 0.4$) (Supplementary Fig. 8). Selected area diffraction patterns document that the material is crystalline and does not show a crystallographic preferred orientation (CPO, Fig. 4b).

Our observations of strong and spatially coherent optical anisotropy are surprising given the grain size is about an order of magnitude smaller than the wavelength of visible light (Fig. 3 and Supplementary Fig. 9). Similar coherent optical anisotropy was observed in nanocrystalline slip zones developed in low-pressure friction experiments conducted with calcite fault gouge[40,41] suggesting that optical anisotropy is common in nanocrystalline fault rocks. While the usual cause of spatially coherent optical anisotropy in microcrystalline materials is a strong CPO, it seems more likely that here the optical anisotropy occurs due to interactions of the nanoparticles with passing light forming optical sub-wavelength grating effects[42].

**Table 1 Summary of the mechanical results of all the experiments.**

| Experiment | t (s) | $P_c$ (MPa) | T (°C) | $\dot{d}$ (mm s⁻¹) | $\dot{\gamma}$ (s⁻¹) | $\dot{\varepsilon}$ (s⁻¹) | $\bar{\sigma}$ (MPa) | $\tau$ (MPa) | $\sigma_n$ (MPa) | $\mu$ | th (mm) |
|---|---|---|---|---|---|---|---|---|---|---|---|
| 030HS | 3132 | 500 | 200 | $5.98 \times 10^{-4}$ | $1.01 \times 10^{-3}$ | $1.16 \times 10^{-3}$ | 1457 | 729 | 1228 | 0.59 | 0.835 |
| 028HS | 3481 | 500 | 300 | $7.68 \times 10^{-4}$ | $1.30 \times 10^{-3}$ | $1.50 \times 10^{-3}$ | 922 | 461 | 961 | 0.48 | 0.486 |
| 282MPa[a] | 1320 | 500 | 300 | $1.28 \times 10^{-3}$ | $1.75 \times 10^{-3}$ | $2.02 \times 10^{-3}$ | 3050 | 1525 | 2263 | 0.67 | 0.778 |
| 034HS | 2321 | 500 | 500 | $7.68 \times 10^{-4}$ | $1.29 \times 10^{-3}$ | $1.49 \times 10^{-3}$ | 451 | 226 | 726 | 0.31 | 0.643 |
| 284MP[a] | 1640 | 500 | 500 | $1.24 \times 10^{-3}$ | $2.03 \times 10^{-3}$ | $2.34 \times 10^{-3}$ | 2434 | 1217 | 2052 | 0.59 | 0.692 |
| 038HS[b] | N/A | 500 | 300 | N/A | N/A | N/A | N/A | N/A | N/A | N/A | 0.839 |
| 042HS[c] | 5518 | 500 | 500 | $8.16 \times 10^{-4}$ | $1.38 \times 10^{-3}$ | $1.59 \times 10^{-3}$ | 515 | 257 | 758 | 0.34 | 0.482 |
| | | | | $1.81 \times 10^{-4}$ | $3.05 \times 10^{-4}$ | $3.52 \times 10^{-4}$ | 287 | 144 | 644 | 0.22 | |
| | | | | $5.23 \times 10^{-5}$ | $8.81 \times 10^{-5}$ | $1.02 \times 10^{-4}$ | 123 | 62 | 562 | 0.11 | |
| | | | | $9.02 \times 10^{-5}$ | $1.52 \times 10^{-4}$ | $1.76 \times 10^{-4}$ | 206 | 103 | 603 | 0.17 | |
| | | | | $2.13 \times 10^{-4}$ | $3.59 \times 10^{-4}$ | $4.15 \times 10^{-4}$ | 428 | 214 | 714 | 0.30 | |
| | | | | $1.25 \times 10^{-3}$ | $2.11 \times 10^{-3}$ | $2.44 \times 10^{-3}$ | 782 | 391 | 891 | 0.44 | |
| 064HS[c] | 4257 | 500 | 300 | $4.60 \times 10^{-4}$ | $7.76 \times 10^{-4}$ | $8.96 \times 10^{-4}$ | 371 | 185 | 687 | 0.27 | 0.668 |
| | | | | $7.74 \times 10^{-5}$ | $1.30 \times 10^{-4}$ | $1.51 \times 10^{-4}$ | 94 | 47 | 548 | 0.09 | |
| | | | | $1.23 \times 10^{-4}$ | $2.08 \times 10^{-4}$ | $2.40 \times 10^{-4}$ | 141 | 70 | 572 | 0.12 | |
| | | | | $1.80 \times 10^{-4}$ | $3.03 \times 10^{-4}$ | $3.50 \times 10^{-4}$ | 200 | 100 | 601 | 0.17 | |
| | | | | $1.39 \times 10^{-4}$ | $2.34 \times 10^{-4}$ | $2.70 \times 10^{-4}$ | 265 | 132 | 633 | 0.21 | |
| | | | | $1.85 \times 10^{-4}$ | $3.12 \times 10^{-4}$ | $3.60 \times 10^{-4}$ | 329 | 164 | 666 | 0.25 | |
| | | | | $4.05 \times 10^{-4}$ | $6.82 \times 10^{-4}$ | $7.88 \times 10^{-4}$ | 539 | 270 | 771 | 0.35 | |
| | | | | $5.92 \times 10^{-4}$ | $9.98 \times 10^{-4}$ | $1.15 \times 10^{-3}$ | 661 | 331 | 832 | 0.40 | |

Variable t deformation time, $P_c$ confining pressure, T temperature, $\dot{d}$ mean vertical piston displacement rate, $\dot{\gamma}$ mean shear strain rate, $\dot{\varepsilon}$ equivalent strain rate, $\bar{\sigma}$ equivalent stress, $\tau$ shear stress, $\sigma_n$ normal stress, $\mu$ friction coefficient, and th thickness of shear zone after deformation.
[a]Values from the previous experiments[30] conducted using an initial grain size of ≤200 μm.
[b]Undeformed sample: the thickness of this shear zone is taken as the initial thickness (th$_0$) of all deformed samples.
[c]Load-stepping experiments.
80-s time intervals are selected at the end of each step to calculate the strain rates.

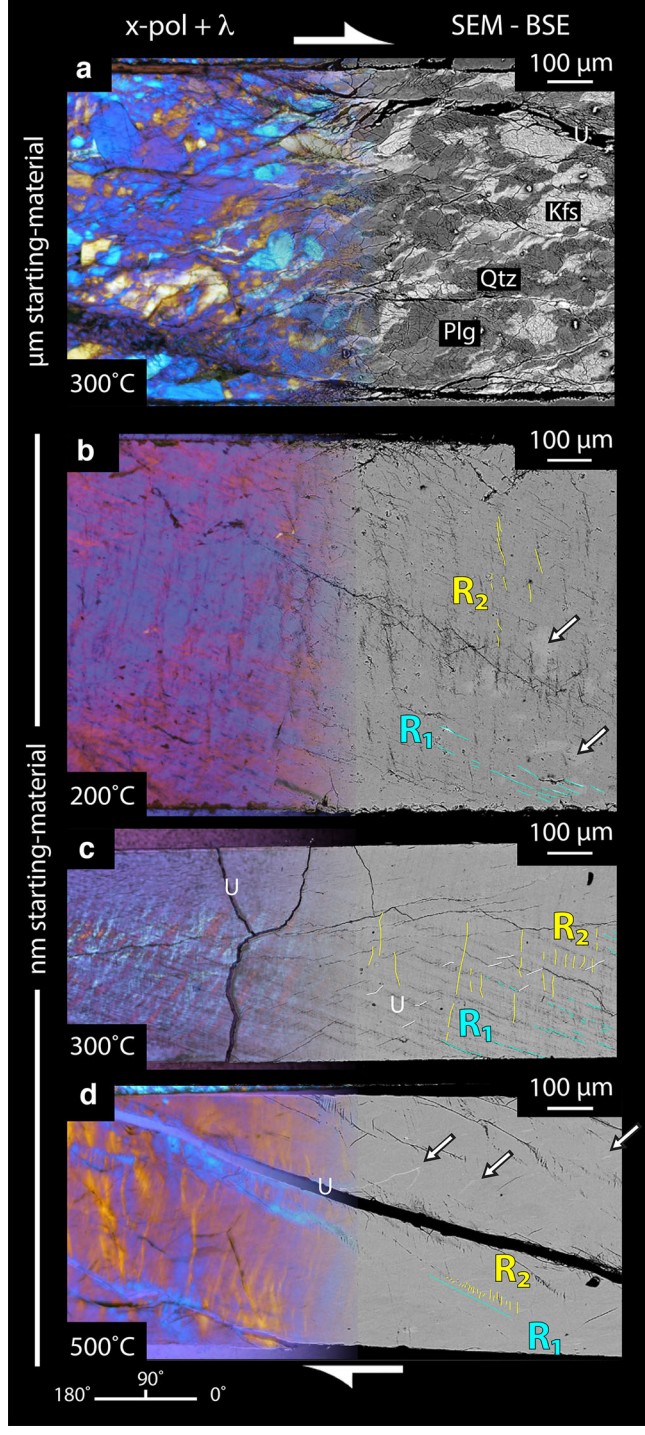

**Fig. 2 Microstructural observations of the sheared fault rocks in polarized light (left) and SEM-BSE (right).** Kfs - potassium feldspar, Qtz - quartz, Plg - plagioclase, U - unloading cracks. $R_1$ & $R_2$ - Riedel shears. Angle convention shown in the lower left. **a** Microcrystalline fault rocks. **b** Nanocrystalline fault rocks deformed at 200 °C. Note pervasive $R_1$ and $R_2$ fractures and locally brighter material in BSE z-contrast indicating different chemical composition and/or density (arrows) **c** Nanocrystalline fault rocks deformed at 300 °C, note alternating cyan/orange layers highlighting kink-bands in the optical image. **d** Nanocrystalline fault rocks deformed at 500 °C. Arrows indicate stretched layers and smeared out domains indicative of continuous flow.

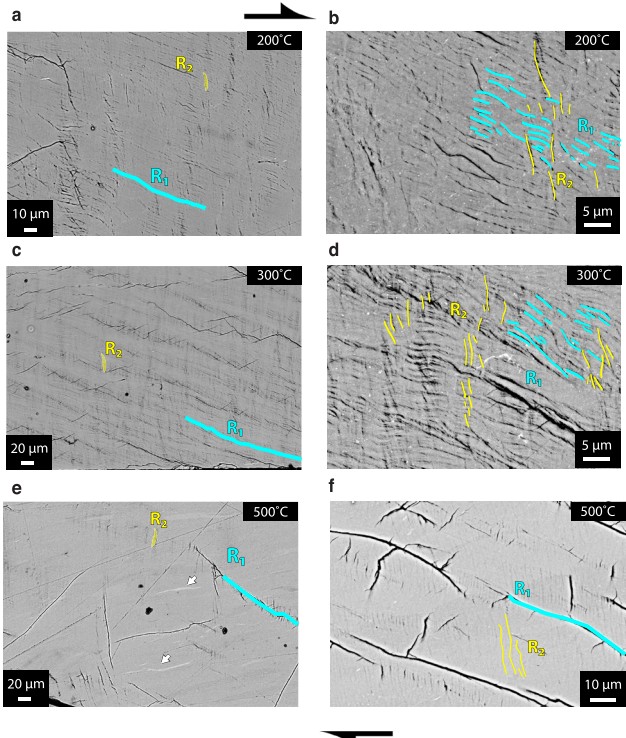

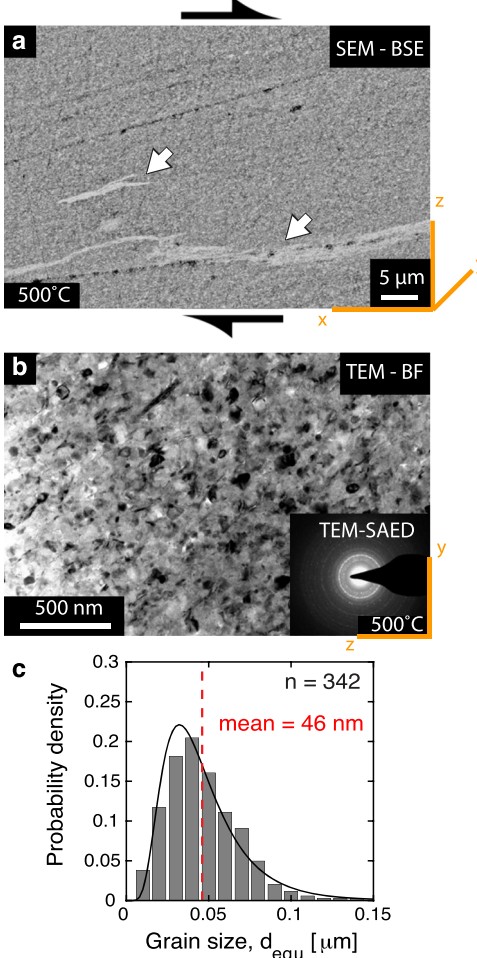

**Fig. 3 Microstructural observations of nanocrystalline fault rocks in BSE.**
**a** $R_1$ oriented fractures (cyan) at 200 °C. Local kinks give rise to $R_2$ orientation (yellow). **b** High-magnification image of microstructure at 200 °C, note pervasive, short $R_1$ oriented fractures and $R_2$ oriented features. **c** Microstructure at 300 °C with pervasive $R_1$ and $R_2$ shears. **d** High-magnification image showing pervasive, short $R_1$ oriented fractures and $R_2$ oriented kink-bands. **e** Microstructure at 500 °C, note smeared out domains of different z-contrast (arrows). $R_1$ fractures form along an array of closely spaced, fine $R_2$ oriented fractures. Note the denser appearance with lower porosity compared to lower T experiments. **f** Detail of en-echelon $R_2$ oriented features.

**Fig. 4 Microstructural observations at high magnifications in SEM and TEM. a** Microstructure at 500 °C, fracture-free portion with folds and stretched layers (white arrows) indicating continuous deformation, i.e., flow. **b** TEM bright-field image of nanocrystalline fault rocks deformed at 500 °C. FIB foil was cut flow perpendicular. Inset shows selected area diffraction pattern (SAED). Notice the equal distribution of diffracting spots suggesting that the material is nanocrystalline with no strong CPO. **c** Grain size distribution of the silicate minerals in (**b**).

## Discussion

In the traditional view of earthquake instability, a decrease in friction coefficient with increasing sliding velocity ("velocity weakening" behavior) is one of the necessary pre-requisites for earthquake nucleation[43]. Although the nanometric flow is believed to play a key role in the earthquake instability[14,26,27], it is difficult to produce "velocity weakening" behavior in viscous fluids especially if they have low-stress exponents. Figure 5a shows the effective stress–strain rate variation at various temperatures calculated using the flow law for nanometric fault rocks. For example, if the strain rate accelerates from $\dot{\bar{\varepsilon}}_0$ to $\dot{\bar{\varepsilon}}_1$, the effective stress will increase as determined by the stress exponent, $n$ ~1.3 (rate-strengthening behavior; Fig. 5a). Theoretically, an increase of temperature due to shear heating could reduce the viscosity of the material, such that the material would be weaker at the higher strain rates (Fig. 5a). However, the low activation energy of the nanomaterial (16,000 ± 14,000 J/mol) precludes an effective temperature-induced weakening effect over a broad range of equivalent stress and ambient temperature conditions. Figure 5b shows the temperature increase (steady-state temperature minus ambient temperature) due to shear heating in the nanocrystalline fault rocks at conditions that could be prevalent at the base of the seismogenic layer (details of the shear heating calculation are in "Methods"). Extremely high equivalent stresses on the order of tens of GPa are required for a modest increase of temperature, meaning that shear-heating instability[6,44] is unlikely

to be the cause of velocity weakening for the nanometric fault rocks. The nanometric fault rocks resist high-velocity deformation and act as viscous breaks. Yet, we observe abrupt failure—a laboratory equivalent of an earthquake—in the microcrystalline experiment at 300 °C where strain localization produces 10–15 vol% of nanocrystalline material[30].

The rheological behavior of the nanocrystalline material suggests a possible weakening mechanism that may lead to an earthquake instability. Fault weakening occurs due to the intrinsic low viscosity of the nanocrystalline fault rocks. Shearing of coarse-grained fault rocks locally leads to comminution and patchy production of nanocrystalline material[22,30], the volume occupied by these nanocrystalline materials increases with increasing work and homologous temperature[24,45], eventually reaching a percolation threshold. Although the extremely fine-grained material is rate-strengthening, it has a much lower viscosity (Figs. 1, 5b) than the surrounding material. Once this material is generated in sufficient quantities and forms a kinematically favorable failure plane, the fault displacement may accelerate on this lower viscosity layer at the same stress level and

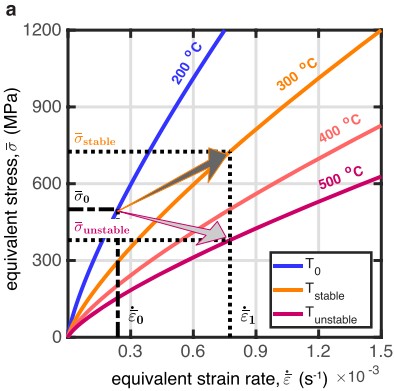
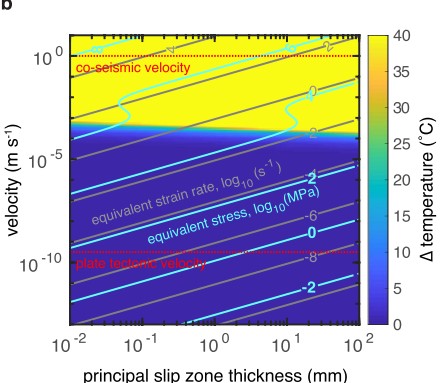

**Fig. 5 Rheology of nanocrystalline fault rocks and implications for earthquake instability. a** Schematic diagram of equivalent stress as a function of equivalent strain rate, as predicted by the experimental flow law. Shear heating can result in "rate weakening" behavior if T increase compensates $\bar{\sigma}$ increase (arrows). **b** Equivalent strain rate and equivalent stress at 300 °C for a range of strain rate values plausible for natural fault zones. Note that the nanocrystalline fault rocks flow at $10^{-4}\,s^{-1}$ at ≈100 MPa stress and at $10^{-6}\,s^{-1}$ at units of MPa, i.e., are extremely weak for the base of the seismogenic zone.

in the absence of significant temperature increase potentially leading to an earthquake instability.

The proposed weakening model here is distinguished from the usual frictional instability but may be analogous to the mechanism causing deep earthquakes, which involve phase transitions[14,15,46], shear-heating induced visco-elastic deformation[6], and "cavitation" based models where a porosity layer coalesces into a failure plane[41]. All these processes have as a common characteristic that there is a weak but rate-strengthening layer. Experiments studying the initialization of deep earthquakes show that shear failure of solid rock at high pressure is possible in the presence of a small amount of low-viscosity material: either a polycrystalline solid that is extremely weak or a true fluid[47]. In our experiments, the nanometric fault rocks deform like a viscous fluid and are extremely weak. This suggests that shearing instability due to weak inclusions may not only be the cause of deep earthquakes but may also plausibly operate in the crust, particularly around the brittle-viscous transition zone where many large earthquakes nucleate.

## Methods

Deformation of granitoid nanocrystalline and microcrystalline fault rocks at seismogenic P-T conditions.

**Starting materials**. The nanomaterials are prepared from crushed Verzasca gneiss powder ($d \leq 200\,\mu m$) of the same origin as the microcrystalline fault rocks described in previous studies[12,30,31]. Image analysis of BSE micrographs shows that the initial powder material consists of 37% quartz (Qtz), 33% plagioclase (Plg), 28% potassium feldspar (Kfs) and minor micas (~2%). Chemical analysis by X-ray fluorescence (XRF) shows that the starting powder consists of 77 wt% $SiO_2$, 13.3 wt% $Al_2O_3$, 4.63 wt% $K_2O$, 3.16 wt% $Na_2O$, and ≤1 wt% of other elements[30].

To grind this material down to nanometric size, the powder is milled for a total of 12 min in deionized water with 0.1 mm diameter zirconia balls using a planetary ball mill (Fritsch Pulverisette 7). After each ~2 min of milling, the holder is left to cool for ~20 min. As the milling is performed in water the temperature increase during milling is buffered at ≤100 °C. The particle size distribution of the unsheared starting material falls in the range $0.01 \leq d \leq 1\,\mu m$, with a median of ~0.1 μm as determined by a laser particle meter (Fritsch Analysette 22). Note that this measurement likely represents the "apparent" grain size as nanoparticles tend to cluster together[48]. After the milling, the powders are dried in an oven at 110 °C for >48 h to evaporate any residual $H_2O$. Supplementary Fig. 1 shows the SEM-BSE and secondary electron (SE) topographic images of the dried nanomaterial. All minerals are mixed due to ball milling so it is impossible to distinguish individual grains. The material forms a compact shard that can be easily disintegrated using a mortar and pestle. Nanomaterials are known to be very reactive, and their properties could change over short periods of time even at room conditions[29,32]. To minimize this inherent uncertainty, the nanomaterial used in this study came from one batch that went through the same history and was used over the period of ≈1 year. The fact that the microstructural as well as mechanical data are consistent between the individual runs indicates that the starting materials at elevated

temperatures and pressures of the experiments have similar properties regardless of the "age" of the starting material.

**Sample assembly preparation**. For each individual experiment performed, we used ~0.1 g room-dry nanomaterial gouge. This gouge is placed between alumina forcing blocks pre-cut at 45°, and weld-sealed in a gold jacket of 0.2 mm wall thickness. The jacket was annealed at 900 °C for two hours in advance to make it malleable. Two alumina pistons are placed on the top and bottom of the jacketed sample to transmit the load from the $\sigma_1$ piston. A 50 mm long graphite furnace around the sample generates heat. Pre-pressed NaCl salt pieces are used as the confining medium. A hole with a diameter of 3.2 mm is drilled in the furnace to place a 2 mm thick alumina ring with inner diameter of 1.5 mm. Another hole with a diameter of 1.6 mm is drilled in the inner salt pieces. A K-type thermocouple goes through these holes and the alumina ring and is directly adjacent to the sample. The sample is oriented to make sure that the thermocouple can measure the temperature in the center of the material. Two 1 mm thick copper discs are placed on the top and bottom of the furnace to allow the current flow through the furnace and heat the sample. The bottom of the assembly is made of a tungsten carbide (WC) plug on which the alumina pistons and inner salt pieces are seated. A pyrophyllite base surrounds the WC plug. Once the sample assembly is placed in the pressure vessel, a piece of lead (Pb) is placed on the top as an upper soft metal disc that transmits the pressure on the sample assembly. See Supplementary Fig. 2 for detailed drawing.

**Experimental apparatus**. A hydraulically driven Griggs-type apparatus is used to conduct the experiments (see ref. [49] for detailed description of the apparatus). The confining pressure applied to the sample assembly is achieved by compressing the salt inside the pressure vessel with $\sigma_3$ piston. The salt sleeves transfer the vertical load applied from the $\sigma_3$ piston into a confining pressure on the sample at the center of the assembly. A differential stress can be applied to the sample with a $\sigma_1$ piston. Both $\sigma_3$ and $\sigma_1$ pistons are controlled by hydraulic rams. The pressure measurements are calculated from the oil pressure within the rams and an external load cell. The temperature of the sample is monitored and controlled by a Euro-therm proportional-integral-derivative (PID) controller connected to the thermocouple.

**Experimental procedures and conditions**. All experiments were conducted at the same confining pressure of 500 MPa and at temperatures of 200, 300, or 500 °C. The whole experiment consists of three stages: pressurization, deformation, quenching and depressurization. First, during the pressurization, we increase the confining pressure and temperature to experimental conditions in 100 MPa and 100 °C increments. Then, in the second stage, two different boundary conditions are used to study the rheological properties of the samples:

- Experiments at constant displacement rate
  In these experiments, the $\sigma_1$ pump is driven at a constant oil flow rate to deform the sample until a desired finite shear strain. The flow rate results in an approximately constant displacement rate of the $\sigma_1$ piston of $10^{-3}\,mm\,s^{-1}$.
- Load-stepping experiments
  In these experiments, the samples are deformed at several load steps. Initially, the $\sigma_1$ piston is driven in at a constant displacement rate until it hits the sample, identically as in the constant displacement rate experiments. After the $\sigma_1$ piston hits the sample, the load is fixed at each step, until the

displacement rate is constant under that load, which indicates that the material is creeping at a steady state (Supplementary Fig. 3).

The last stage of an experiment is quenching and depressurization to bring down the temperature, axial load, and confining pressure to room conditions after deformation. During quenching, the temperature is dropped to the set point (30 °C) automatically and quickly at a rate of 300 °C/min. Simultaneously, the $\sigma_1$ pump is reversely driven to lower the load until the differential stress, $\sigma_1-\sigma_3 \leq 100$ MPa, to minimize unloading cracks. After quenching, both the confining pressure and the load are decreased to room conditions by reducing the oil pressure in the hydraulic rams.

**Data acquisition and processing**. All the experimental variables (temperature, confining pressure, load, axial and $P_c$ piston position, volts and amps in heating circuit, pressures and flow rates in both syringe pumps) are digitally recorded at a frequency of 1 Hz from the beginning of pressurization. The oil pressure in the hydraulic rams is recorded with an external pressure transducer and used to calculate the confining pressure. The force on the sample is measured by an external load cell. The vertical displacement of $\sigma_1$ piston is measured by an external displacement transducer with a max. linearity deviation of 0.13%.

The raw mechanical data are corrected for rig stiffness, total "friction" related to the driving in of the $\sigma_1$ piston into the pressure medium and reducing of contact area between forcing blocks with increasing shear. The stiffness correction accounts for the elastic extension of the machine as load is applied to the sample. Based on the calibration results using an alumina piston, the stiffness of the rig is ~0.0061 mm/kN. "Friction correction" accounts for the internal friction on the ram, the friction on the deformation piston-packing ring, and the friction along different contact surfaces inside the sample assembly. The total friction correction coefficient for the mechanical results in Fig. 1 is 1.31 kN/mm. In addition, an 'area correction' accounts for the decreasing overlap as slip increases along the pre-cut forcing blocks.

The general shear geometry implies that there is a pure shear component in the fault rock. The vertical piston displacement can be decomposed into a thinning component perpendicular to the shear zone and a simple shear component along the shear zone. To calculate the shear strain at any given time, we assume that the thinning rate is constant so that the total amount of thinning is evenly distributed during the deformation. The initial thickness before deformation is determined by the experiment 038HS, which we pressurized to 500 MPa and heated to 300 °C and subsequently quenched.

Table 1 summarizes the mechanical results of all the experiments. Using the nanomaterials, experiments 030HS, 028HS, and 034HS were performed at a constant displacement rate ~$10^{-3}$ mm s$^{-1}$, corresponding to shear strain rate ~$10^{-3}$ s$^{-1}$. Experiments 282MP and 284MP are performed using micropowders at identical conditions[30]. Two load-stepping experiments (042HS and 064HS) were performed to study the stress-strain rate relationship of the material. Table 1 summarizes the results of the six steps in the experiment at 500 °C (042HS) and eight steps in the experiment at 300 °C (064HS). As we described above, converting measured forces and displacements to stresses and strains involves a number of assumptions ("friction correction", piston overlap, partitioning of displacement into thinning and shearing components) and therefore higher-strain data points are associated with a larger error.

**Microstructural observations**. Thin sections are prepared from the samples and the chemical and microstructural properties of the starting as well as deformed materials are investigated in detail via SEM-BSE, optical microscopy, and TEM.

The BSE images are high-resolution compositional maps visualizing the differences in atomic number and density for quickly distinguishing of different phases based on their brightness. For example, phases with greater average atomic number (Z) and/or density show brighter BSE intensity. In contrast, phases with lower average Z and/or density are related to the dark areas. We used a Zeiss Merlin high-resolution scanning electron microscope to obtain the BSE images.

**Estimation of the experimental flow law of nanocrystalline fault rocks**. The rheological properties of a material can be described in terms of a flow law, which relates shear stress ($\tau$) and shear strain rate ($\dot{\gamma}$). In terms of equivalent stress ($\bar{\sigma}$) and equivalent strain rate ($\dot{\bar{\varepsilon}}$), a flow law has the form of

$$\dot{\bar{\varepsilon}} = A\bar{\sigma}^n \exp\left(\frac{-Q}{RT}\right), \tag{1}$$

where $\bar{\sigma} = 2\tau$, and $\dot{\bar{\varepsilon}} = \frac{2\sqrt{3}}{3}\dot{\gamma}$. $A$ is the pre-exponential constant; $n$ the stress exponent (~1 for diffusion creep, 2 for dislocation accommodated grain boundary sliding and ~3–5 for dislocation creep[50]); $R$, the gas constant; $T$, absolute temperature; and $Q$, the activation energy. Notice that the term for grain size dependence in diffusion creep is included in the pre-exponential constant in this formula. This equation can be formulated in terms of logarithms as:

$$\log_{10}\dot{\bar{\varepsilon}} = n\log_{10}\bar{\sigma} + \log_{10}Ae^{-\frac{Q}{RT}}. \tag{2}$$

Therefore, the slope of the $\dot{\bar{\varepsilon}}$ versus $\bar{\sigma}$ curve in the logarithmic coordinate system returns the stress exponent ($n$). By fitting the measured points in Table 1 with a linear model and solving the linear least-square problem, we obtain $n = 1.8 \pm 0.8$ at

500 °C, where the error bars indicate a 95% confidence interval. In the same manner, the stress exponent is $n = 1.0 \pm 0.5$ at 300 °C. $n \approx 1$ indicates that the nanocrystalline fault rocks deform as a linear-viscous (Newtonian) fluid at 300 °C. A higher stress exponent of $n \approx 2$ at 500 °C would be expected if dislocation accommodated grain boundary sliding was the dominant deformation mechanism[51]. However, our 300 °C experiment is better constrained than the 500 °C experiment (Supplementary Fig. 5).

After obtaining the stress exponent $n$, we can determine the activation energy using the constant displacement rate experiments at different temperatures by reformulating Eq. (1) as

$$\log_{10}\bar{\sigma} = \left(\frac{Q}{nR}\log_{10}e\right)\frac{1}{T} + \frac{1}{n}\log_{10}\dot{\bar{\varepsilon}} - \frac{1}{n}\log_{10}A. \tag{3}$$

The slope provides a value for $Q/nR$ and thus determines the activation energy $Q$ of the material because $n$ is determined independently. The estimated activation energy is $Q_{n=1.0} = 12,000 \pm 10,000$ J/mol and $Q_{n=1.8} = 21,000 \pm 18,000$ J/mol by assuming stress exponents of 1.0 and 1.8, respectively, with the caveat that no steady-state stress level was achieved at low temperatures. As can be seen in Eq. (3), the determination of $Q$ in the constant displacement rate experiment is not rigorous since the value of the stress exponent $n$ is convolved. Once $n$ and $Q$ are determined, the pre-exponential constant $A$ can be estimated from the intercept of the $\dot{\bar{\varepsilon}}$ versus $\bar{\sigma}$ curve.

The low activation energy estimated from both experiments 042HS and 064HS indicates that the material is temperature insensitive. Thus, we combine both experiments and determine a unified stress component of $n = 1.3 \pm 0.4$ and activation energy of $Q_{n=1.3} = 16,000 \pm 14,000$ J/mol for the material. In this case, a temperature-independent pre-exponential constant $A$ is estimated from the intercept of the $\log_{10}\bar{\sigma}$ vs. $1/T$ curve with a constant $\dot{\bar{\varepsilon}}$ of $10^{-3}$ s$^{-1}$.

To verify the estimation of the rheological parameters, we calculate the $\dot{\bar{\varepsilon}}$ of experiments 030HS, 028HS, and 034HS using the estimated $n$ and $Q$ and compare them with the experimental values. Supplementary Fig. 4 and Supplementary Table 1 summarize all the calculated $\dot{\bar{\varepsilon}}$ using different pairs of $n$ and $Q$. The calculated $\dot{\bar{\varepsilon}}$ are close to the experimental $\dot{\bar{\varepsilon}}$ of ~$1.0 \times 10^{-3}$ s$^{-1}$. Therefore, our estimation of the rheological parameters is reasonable.

**Sources of error**. All the calculations above use a total friction correction coefficient of 1.31 kN/mm. This value is chosen empirically from a large amount of experimental data (see Fig. 2.27 in ref. [52], see also ref. [53]). The total friction correction coefficient of the rig cannot be accurately measured, but it affects the estimation of the rheological parameters by changing the value of the measured equivalent stress. Supplementary Fig. 5 shows the evaluation of $n$ and $Q$ with different friction correction coefficients. With a range of 0.1–2.2 kN/mm, the estimated $n$ and $Q$ range from 0.9 to 1.4 and from 10,000 to 16,000 J/mol, respectively, using data from both experiments (042HS and 064HS).

**Calculation of the temperature increase by shear heating**. The time-independent temperature increase by shear heating in the nanocrystalline principal slip zone is estimated by solving the steady-state heat equation,

$$-\nabla(k\nabla T) = \frac{Q_{sh}}{\rho C_p}, \tag{4}$$

where $\rho$ is the rock density in kg·m$^{-3}$; $C_p$, the specific heat capacity in J kg$^{-1}$ K$^{-1}$; $k$, thermal conductivity in W m$^{-1}$ K$^{-1}$; $T$, the temperature in Kelvin (K); and $Q_{sh} = \dot{\gamma} \cdot \tau$, the heat generation in J m$^{-3}$ s$^{-1}$. The equation is solved by the finite-difference method to calculate the steady temperature due to shear heating. The boundary condition is the ambient temperature, $T_0$ that is set to the experimental temperature (200, 300 or 500 °C, see Fig. 5b and Supplementary Fig. 6b, c). Only the thickness ($d$) of the principal slip zone needs to be known by solving the one-dimensional heat equation. We assume that the heat source is applied to the middle of the shear zone to estimate the largest possible T increase. We also assume that $k$, $\rho$, and $C_p$ are constant and let $\rho = 2800$ kg·m$^{-3}$, $C_p = 1000$ J kg$^{-1}$ K$^{-1}$, and $k = 2.5$ W m$^{-1}$ K$^{-1}$ (see model parameters for crust in ref. [54]). The shear heating result in Supplementary Fig. 6a is calculated using a thickness of 1 mm, which is close to the initial thickness of the shear zone in our experiments. The equivalent shear strain rate, $\dot{\bar{\varepsilon}}$, is calculated according to the experimental flow law, given $\bar{\sigma}$ ranging from $10^0$ to $10^5$ MPa and $T_0$ ranging from 100 to 1400 °C. Note that the stress and strain rate in Supplementary Fig. 6a is used to calculate the heat source for shear heating rather than the results after shear heating. Extrapolating the experimental data to a range of conditions that are plausible for fault zones at the base of the seismogenic zone we assume that the thickness of the principal slip zone ($d$) ranges from 10 $\mu m$ to 10 cm. The shear velocity, $v$, applied to the boundary of the shear zone ranges from $10^{-13}$ to $10^1$ m/s. Thus, the shear strain rate is given by $\dot{\gamma} = v/d$ and plotted in Fig. 5b and Supplementary Fig. 6b, c. Then the equivalent stress before shear heating can be calculated by the experimental flow law. This equivalent stress for the source of shear heating is not plotted in Fig. 5b and Supplementary Fig. 6b, c. Instead, we plot the contours of the equivalent stresses after accounting for the effect of shear heating.

## Data availability
Source data are provided with this paper. All mechanical data are available on zenodo.org at https://doi.org/10.5281/zenodo.5496000.

## Code availability
Codes for the mechanical data evaluation results in Figs. 1 and 5, and Supplementary Figs. 3–6 can be accessed at https://doi.org/10.5281/zenodo.5496000.

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

## Acknowledgements

Holger Stünitz and Oliver Jagoutz are gratefully acknowledged for copy-editing of an earlier version of this manuscript. Shiahn Chen and Yong Zhang are thanked for assistance with FIB and TEM imaging. Funding by NSF EAR-1833478 & EAR-2054414 for laboratory technician support are gratefully acknowledged. This work made use of the MRSEC Shared Experimental Facilities at MIT, supported by the National Science Foundation under award number DMR-1419807. H.S. was supported by MIT Earth Resources Laboratory and MIT MathWorks Science Fellowship. Funding for the article-processing charge was partially supplied by the MIT Libraries.

## Author contributions

H.S. performed the experiments on nanocrystalline rocks, evaluated the mechanical and microstructural data. M.P. designed the study, performed the experiments on microcrystalline rocks, and performed quantitative image analysis. Both authors participated in the interpretation of the data and co-wrote the text.

## Competing interests

The authors declare no competing interests.

## Additional information

**Peer-review information** *Nature Communications* thanks the anonymous reviewer(s) for their contribution to the peer review of this work. Peer reviewer reports are available.

