## [Peer Review File · Nature Communications]

REVIEWER COMMENTS

Reviewer #2 (Remarks to the Author):

This paper reports experimental results of shear in granitoid nanograin gouge, and compares them to similar experiments using micro-grain gouge, under ambient temperature of 200-500C and confining stress of 500MPa, conditions typical of the base of the brittle ductile transition (BDT). As far as I know, experiments quantifying mechanical properties of bulk nanograin samples of rocks were not attempted before, though nanograins are often observed to line experimental and natural fault-zones.

The results of the experiments are striking, though perhaps not surprising in light of previous experiments where nano-grain formation was observed: nanograin samples are observed to be much weaker than micro-grain samples. They shear at 1/5-1/4 of the strength of micro-grains. In addition, they show velocity strengthening behaviour and a very low activation energy, that is 1/4- 1/3 the value in intact granite. I think the results are groundbreaking for the field of rock and earthquake mechanics, especially in terms of consequences for earthquake initiation at the BDT. The paper suggests that shearing micro-grains will progressively form patches of weak, rate-strengthening, thermally insensitive nanograins. When these patches coalesce to form a percolating weak network, there will an unstable collapse of the system, which is basically a "one-time" failure or earthquake. This is as opposed to the rate and state earthquakes that occur repeatedly on the same fault, and are the usual description of earthquakes.

I suggest publication in NC, as this is a very interesting & important work, that will deeply influence the way we understand earthquakes at the BDT.

However there are some issues that need to be addressed before publication:

some figures need clarification: It is not clear for Figure 1c which experiments are shown. I think it shows 2 experiments, in 2 different colored symbols, but this is not explained, I merely deduced that. Please add in the caption/text which experiments are these? which Table shows experimental parameters? E.g. What temperature is each one? add legend? And what is the equivalent stress sigma in the x-axis, I didn't see where it is defined. I mean, in fig 1a they measure tau, the shear stress, but where is the equivalent stress sigma defined, and from which experimental figure is it measured? In fig 1b- please state in caption or text, how is viscosity measured? From which experiments and graphs and by what technique?

In eqn 1 I suggest to use consistent units of J (not kJ and J) for activation energy and R.

missing symbols in line 421, 424,440.

Reviewer #3 (Remarks to the Author):

The paper addresses the issue of the controlling mechanism of earthquakes, which is a central and puzzling topic in geosciences. It is widely recognized that slip along active fault-zones is localized within a thin zone that is composed of nano-scale powder. The authors focus on understanding the enigmatic rheology of this powder and its effect on earthquake instability. Overall, this is a very good paper on an important and timely topic that deserves to be published in a journal like Nature Communications. Yet, it is suggested that before accepted for publication, the authors will work on a few issue outlined below. The strong points of the paper are listed first, followed by several major comments, which are mostly related to completeness of background and discussion, and multiple minor editorial comments.

First, the analysis presents unique observations because conducting high-pressure/high-temperature experiments with nano-materials is a challenging task that the authors performed excellently. The section on Methods is an outstanding example of how to describe the essential details of an elaborate

laboratory work and how to analyze the observations. This type of analysis will serve as a basis for future works.

Major comments:

1. The experimental design is limited to small displacements and low velocities as imposed by high-pressure/high-temperature experiments. However, many long- displacement experiments show weakening after a period of strengthening. For example, the authors referred to the "viscous braking" phenomenon, which is also observed in experiments focused on fault melting, e.g., Hirose & Shimamoto, 2005. Similar transition from strengthening to weakening cannot be detected in the present setting. It is unknown if nanopowder will show such behavior, but the authors recognize and accept this limitation.

2. The nanomaterial was pre-prepared before the experimental shearing for obvious practical reasons. However, nano-scale materials are very reactive due to the large surface areas and frequently they form aggregates. The properties of powders formed during fault slip may have a short "life" and they may strengthen even during shear, e.g., Reches and Lockner, 2010. This may not be a problem in the present experiments, yet the authors need to mention the possible difference between "fresh" and "pre-fabricated" powders. It is anticipated that mechanical evolution of the powder was very active here due to high PT and long duration of the experiments. Admittedly, this effect of time-dependent strengthening is poorly understood in general, and reaches beyond the present paper.

3. The authors emphasize the distinction between rate-state friction and the results of their analyses (abstract and discussion). While this deduction is correct, it is suggested to de-emphasize this secondary effect of contradicting the RS friction. Instead, in the discussion, the authors present a preferred application related to creep and nucleation that should be emphasized as the main potential application.

4. The concept of powder lubrication as a weakening mechanism was mentioned in high-velocity experiments (references below including engineering literature). While the present experimental approach is different, the objective is similar: mechanisms of dynamic fault weakening. The authors need to clarify the approaches differences and their importance, if any.

5. What is the material under investigation here is: nanopowder, nanometric, nanomaterial, nanocrystalline? Are all these the same? Either explain the differences or revise to remove confusion. For example, crystalline material may differ mechanically from amorphous material. It is true that at the nano-scale the distinction may be difficult.

Hirose, T., & Shimamoto, T. (2005). Slip-weakening distance of faults during frictional melting as inferred from experimental and natural pseudotachylytes. *BSSA*, 1666-1673.

Reches, Z. E., & Lockner, D. A. (2010). Fault weakening and earthquake instability by powder lubrication. *Nature*, 452-455.

Han, R., Hirose, T., & Shimamoto, T. (2010). Strong velocity weakening and powder lubrication of simulated carbonate faults at seismic slip rates. *JGR* 115.

Wilson, B., et al. (2005). "Particle size and energetics of gouge from earthquake rupture zones." *Nature* 749-752.

Heshmat, H. (1991). The rheology and hydrodynamics of dry powder lubrication. *Tribology Transactions*, 34, 433-439.

Minor comments (listed by line #)

6 Replace 'Unfortunately' by 'However'

12 The very low exponent and activation energy are interesting and puzzling. There is no attempt to explain them and show other cases.

14-16 the statement "... the spatial extent and topology of the weak layers and the amount of stored elastic energy in the surrounding wall rock." is always correct and thus redundant.

40 what is the mechanical difference between amorphous and crystalline? aren't they all grains?

44 delete 'by viscous processes'
49-50 good questions! Only partly answered.
59 "...sticks together.." is poorly defined terms, be specific or delete
61 can you refer to 'true' vs 'apparent' grain size, which is an old problem, e.g., Wilson et al., 2005)
69 replace 200-500 by '200, 300, 500'
83 why is the statement "... just below the Goetze criterion" important?
87 replace 'Comparison ...' by 'The shear strength of...'
112 it is not clear why these observations suggest 'diffusion creep'. give references to demonstrate.
113-114 give references for such '..microcrystalline materials...'
124, 130, 134, 135, 145 note typos
142 '..folds..' are not necessary evidence for flow
Fig. 2C what are the N-S orange zones in left side of (c)? R2? kinks?
174 No folds could be seen; what are the 'layers' that are being stretched?
175 Fig. 3C displays granular material with no flow structures
184-186 this is probably the most central deduction from the experimental observations. Somehow it is hidden.
197 which 'stresses' shear or normal?
199 replace ; by period .
200 it is correct that one stick-slip was observed, yet in typical stick-slip experiments there are multiple events like this one. So just one thus not necessarily implies a characteristic mode.
201-202 this point will become a major argument later. How well was it documented in the present experiments?
203-204, and 215-216 delete reference to RS friction
209-213 this central argument is stated in general, plausible terms. It could be further strengthened with additional hard data from the experiments.
234 the mentioned stress levels are "weak" only for a deep fault!
421, 424, 440 typos
506 in their careful analysis, the authors demonstrated the viscous properties of the nanopowder. They should compare these properties to other materials, natural rocks (e.g., talc, shales or salt) and/or industrial (e.g., molten tar etc)

Reply to Reviewers' comments

We thank the reviewers for their assessment of our manuscript entitled " Nanometric flow and earthquake instability". We have revised the manuscript based on the constructive comments made by the reviewers, including rewriting / reorganizing the main text and figures. Below we specify how we have improved the manuscript, addressing each of the points made by the reviewers.

REVIEWER COMMENTS

Reviewer #2 (Remarks to the Author):

This paper reports experimental results of shear in granitoid nanograin gouge, and compares them to similar experiments using micro-grain gouge, under ambient temperature of 200-500C and confining stress of 500MPa, conditions typical of the base of the brittle ductile transition (BDT). As far as I know, experiments quantifying mechanical properties of bulk nanograin samples of rocks were not attempted before, though nanograins are often observed to line experimental and natural fault-zones.

The results of the experiments are striking, though perhaps not surprising in light of previous experiments where nano-grain formation was observed: nanograin samples are observed to be much weaker than micro-grain samples. They shear at 1/5-1/4 of the strength of micro-grains. In addition, they show velocity strengthening behaviour and a very low activation energy, that is 1/4-1/3 the value in intact granite. I think the results are groundbreaking for the field of rock and earthquake mechanics, especially in terms of consequences for earthquake initiation at the BDT. The paper suggests that shearing micro-grains will progressively form patches of weak, rate-strengthening, thermally insensitive nanograins. When these patches coalesce to form a percolating weak network, there will an unstable collapse of the system, which is basically a "one-time" failure or earthquake. This is as opposed to the rate and state earthquakes that occur repeatedly on the same fault, and are the usual description of earthquakes.

I suggest publication in NC, as this is a very interesting & important work, that will deeply influence the way we understand earthquakes at the BDT.

We thank the reviewer for recognizing the impact of our work and its suitability for publication in Nature Communications.

However there are some issues that need to be addressed before publication:
some figures need clarification: It is not clear for Figure 1c which experiments are shown. I think it shows 2 experiments, in 2 different colored symbols, but this is not explained, I merely deduced that. Please add in the caption/text which experiments are these?

The experiment labels (042HS and 064HS) have been added to the caption of Fig 1c. and we have referenced the stress-strain rate curves of these experiments in Supplementary Fig. 3. We have also improved the legend of figure 1 to provide more clarity.

which Table shows experimental parameters? E.g. What temperature is each one? add legend?

The “Extended Data Table 1” in the original submission shows the experimental parameters of all the constant displacement rate experiments and “Supplementary Data Table 1” was showing all the individual load steps for the load stepping experiments. We have now combined these two tables into one for reader convenience. Now with the new format, the table become Table 1.

And what is the equivalent stress σ in the x-axis, I didn't see where it is defined. I mean, in fig 1a they measure τ , the shear stress, but where is the equivalent stress σ defined, and from which experimental figure is it measured?

The equivalent stress $\bar{\sigma}$ is defined as $\bar{\sigma} = 2\tau$ where τ is the shear stress. Details about this definition and calculations in Fig 1c can be found in the section of “Methods” (subsection: Estimation of the experimental flow law of nanocrystalline fault rocks). We added a reference to the figure legend to the “Methods” section with the definitions.

In fig 1b- please state in caption or text, how is viscosity measured? From which experiments and graphs and by what technique?

We use the definition of apparent viscosity in Fig. 1b which is calculated by the ratio between shear stress and shear strain rate. The mechanical data plotted with the same shape of line in both Fig. 1a and Fig. 1b are from the same experiments. We have added that it is an “apparent” viscosity and improved the legend of Figure 1b for clarity.

In eqn 1 I suggest to use consistent units of J (not kJ and J) for activation energy and R.

Sure, point taken.

missing symbols in line 421, 424,440.

Sorry for these typos, we fixed them. They are probably due to the online word to pdf conversion in the manuscript submission system.

Reviewer #3 (Remarks to the Author):

Dear Editor and Authors,

The paper addresses the issue of the controlling mechanism of earthquakes, which is a central and puzzling topic in geosciences. It is widely recognized that slip along active fault-zones is localized within a thin zone that is composed of nano-scale powder. The authors focus on understanding the enigmatic rheology of this powder and its effect on earthquake instability. Overall, this is a very good paper on an important and timely topic that deserves to be published in a journal like Nature Communications. Yet, it is suggested that before accepted for publication, the authors will work on a few issue outlined below. The strong points of the paper are listed first, followed by several major comments, which are mostly related to completeness of background and discussion, and multiple minor editorial comments.

First, the analysis presents unique observations because conducting high-pressure/high-temperature experiments with nano-materials is a challenging task that the authors performed excellently. The section on Methods is an outstanding example of how to describe the essential details of an elaborate laboratory work and how to analyze the observations. This type of analysis will serve as a basis for future works.

Thank you very much for your constructive comments and your agreement on the importance of our discovery.

Major comments:

1. The experimental design is limited to small displacements and low velocities as imposed by high-pressure/high-temperature experiments. However, many long- displacement experiments show weakening after a period of strengthening. For example, the authors referred to the “viscous braking” phenomenon, which is also observed in experiments focused on fault melting, e.g., Hirose & Shimamoto, 2005. Similar transition from strengthening to weakening cannot be detected in the present setting. It is unknown if nanopowder will show such behavior, but the authors recognize and accept this limitation.

Yes, we agree with this statement completely. We have focused exclusively on slow-slip velocity experiments as we are interested in the process of nucleation in this paper. We have added text on Lines 46-48 to clarify the connection to high-velocity experiments in line with point 4 below.

2. The nanomaterial was pre-prepared before the experimental shearing for obvious practical reasons. However, nano-scale materials are very reactive due to the large surface areas and frequently they form aggregates. The properties of powders formed during fault slip may have a short “life” and they may strengthen even during shear, e.g., Reches and Lockner, 2010. This may not be a problem in the present experiments, yet the authors need to mention the possible difference between “fresh” and “pre-fabricated” powders. It is anticipated that mechanical evolution of the powder was very active here due to high PT and long duration of the experiments. Admittedly, this effect of time-dependent strengthening is poorly understood in general, and reaches beyond the present paper.

We agree. The powders are certainly very reactive, and their properties could change over short periods of time even at room conditions. To minimize this inherent uncertainty, the nanopowder used in this study came from one batch that went through the same history. The fact that the microstructural as well as mechanical data are consistent between the individual runs indicates, that either the microstructural changes occur over longer time periods or that the powders at elevated temperatures and pressures of the experiments have similar properties. We extended the discussion on lines 281-287 to cover this point.

3. The authors emphasize the distinction between rate-state friction and the results of their analyses (abstract and discussion). While this deduction is correct, it is suggested to de-emphasize this secondary effect of contradicting the RS friction. Instead, in the discussion, the authors present a preferred application related to creep and nucleation that should be emphasized as the main potential application.

Ok, thank you for the suggestion. We have slightly modified the text to de-emphasize the contradiction, while still stating it as it is an important deduction in our opinion that is better stated up-front than left for the readers to figure out.

4. The concept of powder lubrication as a weakening mechanism was mentioned in high-velocity experiments (references below including engineering literature). While the present experimental approach is different, the objective is similar: mechanisms of dynamic fault weakening. The authors need to clarify the approaches differences and their importance, if any.

We expanded on this point on lines 46-48. The main difference is that in our experiments we are isolating the effect of heating as we perform the experiments under slow displacement rates and at controlled and constant P and T conditions and determine the intrinsic properties of the nanomaterials.

5. What is the material under investigation here is: nanopowder, nanometric, nanomaterial, nanocrystalline? Are all these the same? Either explain the differences or revise to remove confusion. For example, crystalline material may differ mechanically from amorphous material. It is true that at the nano-scale the distinction may be difficult.

We simplified our use of terminology. We use now throughout the manuscript either nanomaterial as the broadest category or nanocrystalline fault rock for the ball-milled material at high pressure and temperature conditions. Nanometric is just a distinction of length scale.

Hirose, T., & Shimamoto, T. (2005). Slip-weakening distance of faults during frictional melting as inferred from experimental and natural pseudotachylytes. *BSSA*, 1666-1673.

Reches, Z. E., & Lockner, D. A. (2010). Fault weakening and earthquake instability by powder lubrication. *Nature*, 452-455.

Han, R., Hirose, T., & Shimamoto, T. (2010). Strong velocity weakening and powder lubrication of simulated carbonate faults at seismic slip rates. *JGR* 115.

Wilson, B., et al. (2005). "Particle size and energetics of gouge from earthquake rupture zones." *Nature* 749-752.

Heshmat, H. (1991). The rheology and hydrodynamics of dry powder lubrication. *Tribology Transactions*, 34, 433-439.

We added the relevant references.

Minor comments (listed by line #)

6 "*Unfortunately, the rheology of nanocrystalline fault rocks remains poorly constrained.*"

Replace 'Unfortunately' by 'However'

Fixed, thanks for pointing this out

12 . "*However, the low measured stress exponent, n , of 1.3 ± 0.4 and the low activation energy, Q , of 16 ± 14 kJ/mol imply that the material will be strongly rate strengthening with a weak temperature sensitivity.*"

The very low exponent and activation energy are interesting and puzzling. There is no attempt to explain them and show other cases.

The low stress exponent is not surprising given that the rocks are inferred to deform by diffusion creep. We have added a classic reference deriving the characteristic stress exponent of 1 for diffusion creep. What is unusual is the low experimental temperature at which diffusion creep occurs. This is due to the very short diffusive distances in the nanomaterial as we discuss together with the low activation energy origins on Lines 113-120 where we also provided references to papers that deal exclusively with the topic.

14-16 “*Whether an earthquake instability develops in fault rocks where nanocrystalline materials form in-situ is dependent upon the spatial extent and topology of the weak layers and the amount of stored elastic energy in the surrounding wall rock.*”

the statement “... the spatial extent and topology of the weak layers and the amount of stored elastic energy in the surrounding wall rock.” is always correct and thus redundant.

We have re-worked the abstract so that it fits with journal requirements, we addressed these concerns in the rewrite. While in principle we agree with the reviewer that the amount of stored elastic energy and spatial extent and topology will always control failure, we find this an important point that bears repeating for clarity for a broad audience.

40 “*Nanocrystalline fault rocks typically form compact zones with little to no porosity, composed of crystals (1’s - 10s of nm) to crystal aggregates (100’s of nm) that are sometimes embedded in an amorphous matrix⁴⁻¹¹. The small grain size and associated short diffusional distances in combination with highly disordered lattices make diffusion competitive at conditions where cataclastic flow typically dominates^{11,30}.*”

what is the mechanical difference between amorphous and crystalline? aren't they all grains?

The mechanical behaviors of amorphous and crystalline materials is different due to the differing defects that accommodate deformation. They can be all grains, but the grains are ordered irregularly for amorphous but regularly for crystalline lattice. Amorphous materials also form matrix in which nanocrystals are embedded which defies the usual meaning of a “grain”.

44 “*Numerous lines of evidence suggest that the nanocrystalline fault rocks are weaker than the surrounding, coarser-grained, material and flow by viscous processes even at low ambient temperatures^{5,11}”*

delete ‘by viscous processes’

Ok, we can bring this point later on lines 241 -242.

49-50 “*at what stage of fault slip did the material form? Did the material reach high temperatures during slip? Is the nanocrystalline material the cause or the consequence of failure?*”

good questions! Only partly answered.

We find that for our studied scenario we did answer these questions 1) prior to failure 2) no 3) cause. We clarified the text to bring this point over more clearly.

59 *“The material sticks together in a dense shard of uniform gray in BSE images suggesting a uniform chemical composition.”*

“..sticks together..’ is poorly defined terms, be specific or delete

Ok, we changed to “forms a cohesive material”

61 *“Initial grain size determined by laser particle sizer is $0.01 \leq d \leq 1$ nm, with a median of ~ 100 nm.”*

can you refer to ‘true’ vs ‘apparent’ grain size, which is an old problem, e.g., Wilson et al., 2005)

Ok, we added this discussion and cited the reference.

69 replace 200-500 by ‘200, 300, 500’

Fixed, thanks for pointing this out

83 *“Only for the nanocrystalline fault rocks, the differential stress, $\Delta\bar{\sigma}$, is just below the Goetze criterion $\Delta\bar{\sigma} \leq Pc$.”*

why is the statement “... just below the Goetze criterion” important?

Here it is stated in the results section as an important observation that gets interpreted and discussed in the discussion on lines 152-159. The nanopowders creep at 500°C in our experiments with little fracturing and the measured differential stress $\Delta\bar{\sigma}$ is smaller than the Pc suggesting that volume conserving flow is possible.

87 *“Comparison of the strength between nanopowders $\sim 0.1 \mu\text{m}$ and micropowders $\leq 200 \mu\text{m}$ ”*
replace ‘Comparison ...’ by ‘The shear strength of...’

Thank you, we changed Comparison of the strength -> Comparison of the shear strength

112 *“Hence, it appears that nanometric fault rocks can dominantly deform by diffusion creep even at low temperatures and fast experimental strain rates.”*

it is not clear why these observations suggest ‘diffusion creep’.

the stress exponent of 1 is indicative of diffusion creep, we have added a reference to a classical paper that establishes this relationship and clarified this inference.

113-114 *“While the stress exponent is consistent with microcrystalline materials deforming dominantly by diffusion creep, the activation energy is substantially lower in the nanocrystalline fault rocks.”*

give references for such ‘..microcrystalline materials...’

all microcrystalline materials deforming by diffusion creep have a stress exponent of 1 as can be demonstrated theoretically and discussed in detail in Raj and Ashby 1971 for example. We have made this point clearer.

124, 130, 134, 135, 145 note typos

Sorry for these typos. They are probably due to the online word-to-pdf conversion program in the manuscript submission system.

142 “*Flow features, such as folds and smeared out domains of slightly different z-contrast, dominate (Fig. 2c, 3b and Extended Data 144 Fig. 7).*”

‘..folds..’ are not necessary evidence for flow

True, but they are evidence of continuous deformation on the observation scale. We clarified this point.

Fig. 2C what are the N-S orange zones in left side of (c)? R2? kinks?

Yes these are kinks that form in the R2 orientation. We have moved and modified Extended Data Figure 9 and moved it into the main text to better show the microstructures of the experiments.

174 “*Fig. 3 ... b, Microstructure at 500°C, fracture-free portion with folds and stretched layers (white arrows) indicating viscous flow.*”

No folds could be seen; what are the ‘layers’ that are being stretched?

Ok, we re-phrased.

175 “*Fig. 3 ... c, TEM bright field image of nanocrystalline fault rocks deformed at 500°C. FIB foil was cut flow perpendicular.*”

Fig. 3C displays granular material with no flow structures

But the SEM images document in Fig 2d and 3b show stretched layers indicative of continuous flow. Furthermore, there is almost no porosity in the material indicating that diffusive mass transfer must be occurring to eliminate porosity as it would form during granular flow. At last the low stress exponent around 1 supports this interpretation as mentioned above - the confluence of these microstructural and mechanical data is a strong argument for this statement in our opinion. We clarified this point in the text.

184-186 “Although the nanometric flow is believed to play a key role in the earthquake instability¹⁰⁻¹², it is difficult to produce “velocity weakening” behavior in viscous fluids especially if they have low stress exponents.”

this is probably the most central deduction from the experimental observations. Somehow it is hidden.

Ok, thank you for the suggestion - we have pointed this observation out in the re-worked abstract to give it more prominence.

197 “Extremely high equivalent stresses on the order of tens of GPa are required for a modest increase of temperature, meaning that shear-heating instability^{20,38} is unlikely to be the cause of velocity weakening for the nanometric fault rocks; the nanometric fault rocks resist high-velocity deformation and act as viscous breaks.”

which ‘stresses’ shear or normal?

They are equivalent stress, $\bar{\sigma}$ calculated as $\bar{\sigma} = 2\tau$ where τ is shear stress. We elaborate on the conversion from shear stresses and strains to equivalent stresses and strains in the Methods section and state that in the text.

replace ; by period .

Fixed, thanks for pointing this out

200-202 “Yet, we observe abrupt failure - a laboratory equivalent of an earthquake - in the microcrystalline experiment at 300°C where strain localization produces 10-15 vol% of nanocrystalline material⁸.”

200 it is correct that one stick-slip was observed, yet in typical stick-slip experiments there are multiple events like this one. So just one thus not necessarily implies a characteristic mode.

To see multiple events in stick-slip experiments, the displacement of the sigma 1 piston should be continued; however, this experiment is ending after we see the failure, so we did record multiple events. We did however another experiment at identical conditions where the same abrupt failure can be observed as reported in Pec et al. 2016, we included just one experiment to avoid making the figures too busy.

201-202 this point will become a major argument later. How well was it documented in the present experiments?

These experiments are mainly documented in Pec et al. 2016 and Pec and Al Nasser 2021 in detail, so here we just briefly mention its difference compared with the nanopowders. This value is established based on tens of experiments (see figure from Pec and Al Nasser 2021 below which shows the volume% of slip zone material as a function of strain energy density. Each dot represents one experiment.)

203-205 “*The rheological behavior of the nanocrystalline material suggests a possible weakening mechanism conceptually different from the existing rate-and-state friction model that may nevertheless lead to an earthquake instability.*”

215-216 “*The proposed weakening model here is distinguished from the usual rate-and-state frictional instability but may be analogous to the mechanism causing deep earthquakes*”

203-204, and 215-216 delete reference to RS friction

Ok, we slightly modified the text. Rate-and-state friction is the commonly used model so we find it important to compare and contrast our model.

209-213 “*Although the extremely fine-grained material is rate strengthening, it has a much lower viscosity (Fig. 1, 4b) than the surrounding material. Once this material is generated in sufficient quantities and forms a kinematically favorable failure plane, the fault displacement may accelerate on this lower viscosity layer at the same stress level and in the absence of significant temperature increase potentially leading to an earthquake instability.*”

this central argument is stated in general, plausible terms. It could be further strengthened with additional hard data from the experiments.

This is our current conclusion based on the confluence of experimental observations. As the data derived from the experiments are fundamentally assuming a steady-state, more complex numerical models will be needed to fully explore the slip behavior in such scenario. The stated inference is a plausible scenario based on our current state in research.

234 “Fig 4 caption: ... Note that the nanocrystalline fault rocks flow at 10^{-4} s⁻¹ at ≈ 100 MPa stress, i.e. are extremely weak.”

the mentioned stress levels are “weak’ only for a deep fault!

True, we have clarified that 100 MPa stress is extremely weak at the base of the seismogenic layer and added that the rocks would be much weaker at lower strain rates.

421, 424, 440 typos

Sorry for these typos. They are probably due to the online word to pdf conversion in the manuscript submission system.

506 “*Extended Data Fig. 4 and Supplementary Data Table 2 summarizes all the calculated $\dot{\epsilon}^-$ using different pairs of n and Q . The calculated $\dot{\epsilon}^-$ are close to the experimental $\dot{\epsilon}^-$ of $\sim 1.0 \times 10^{-3}$ s⁻¹. Therefore, our estimation of the rheological parameters is reasonable.*”
in their careful analysis, the authors demonstrated the viscous properties of the nanopowder. They should compare these properties to other materials, natural rocks (e.g., talc, shales or salt) and/or industrial (e.g., molten tar etc)

We prefer to keep as it is as it relates to the validation of our flow law.

REVIEWERS' COMMENTS

Reviewer #2 (Remarks to the Author):

All my concerns have been answered in the revised version. This is a very good paper now and should be published.

Reviewer #3 (Remarks to the Author):

The authors did very good job in revising the article. No further comments are needed and the paper is ready for publication.